# Effects of T cell leptin signaling on systemic glucose tolerance and T cell responses in obesity

**Kaitlin Kiernan[1], Amanda G. Nichols[2], Yazan Alwarawrah[2], Nancie J. MacIver [2,3]***

**1** Department of Immunology, Duke University School of Medicine, Durham, North Carolina, United States of America, **2** Department of Pediatrics, University of North Carolina School of Medicine, Chapel Hill, North Carolina, United States of America, **3** Department of Nutrition, University of North Carolina School of Medicine and Gillings School of Global Public Health, Chapel Hill, North Carolina, United States of America

\* nancie_maciver@med.unc.edu

**Data Availability Statement:** All relevant data are within the paper and its Supporting information files.

**Funding:** This study was supported by NIH R01-DK106090 (NJM) The funders had no role in study

## Abstract

### Background/Objectives

Leptin is an adipokine secreted in proportion to adipocyte mass and is therefore increased in obesity. Leptin signaling has been shown to directly promote inflammatory T helper 1 (Th1) and T helper 17 (Th17) cell number and function. Since T cells have a critical role in driving inflammation and systemic glucose intolerance in obesity, we sought to determine the role of leptin signaling in this context.

### Methods

Male and female T cell-specific leptin receptor knockout mice and littermate controls were placed on low-fat diet or high-fat diet to induce obesity for 18 weeks. Weight gain, serum glucose levels, systemic glucose tolerance, T cell metabolism, and T cell differentiation and cytokine production were examined.

### Results

In both male and female mice, T cell-specific leptin receptor deficiency did not reverse impaired glucose tolerance in obesity, although it did prevent impaired fasting glucose levels in obese mice compared to littermate controls, in a sex dependent manner. Despite these minimal effects on systemic metabolism, T cell-specific leptin signaling was required for changes in T cell metabolism, differentiation, and cytokine production observed in mice fed high-fat diet compared to low-fat diet. Specifically, we observed increased T cell oxidative metabolism, increased CD4+ T cell IFN-γ expression, and increased proportion of T regulatory (Treg) cells in control mice fed high-fat diet compared to low-fat diet, which were not observed in the leptin receptor conditional knockout mice, suggesting that leptin receptor signaling is required for some of the inflammatory changes observed in T cells in obesity.

### Conclusions

T cell-specific deficiency of leptin signaling alters T cell metabolism and function in obesity but has minimal effects on obesity-associated systemic metabolism. These results suggest

design, data collection and analysis, decision to publish, or preparation of the manuscript.

**Competing interests:** The authors have declared that no competing interests exist.

a redundancy in cytokine receptor signaling pathways in response to inflammatory signals in obesity.

## Introduction

Obesity, which is increasing at an alarming rate in the United States and other developed countries [1], is associated with a wide range of comorbidities, including type 2 diabetes, hyperlipidemia, hypertension, heart disease, ischemic stroke and several types of cancer [2–7]. Numerous studies have demonstrated that the development of many of these obesity-associated comorbidities is promoted by a chronic, low grade inflammatory state [8–12]. In mouse studies of visceral adipose tissue, pro-inflammatory immune cells are preferentially increased in the adipose tissue of obese mice compared to lean mice. In particular, increases in M1-like pro-inflammatory macrophages, CD8$^+$ T cells, T helper 1 (Th1) cells, and B cells are observed [13]. In contrast, there are decreased percentages of T regulatory (Treg) cells, T helper 2 (Th2) cells, natural killer T (NKT) cells, eosinophils, and type 2 innate lymphoid (ILC2) cells. This influx of inflammatory immune cells into obese adipose tissue promotes the production of inflammatory cytokines such as tumor necrosis factor (TNF), interleukin-6 (IL-6), and interferon gamma (IFN-γ), which contribute to the chronic inflammatory state (metaflammation) observed in obesity [13–19]. Moreover, inflammatory cytokine production by adipose tissue immune cells has been shown to induce insulin resistance in obesity.

Although pro-inflammatory macrophages are the most predominant immune cell population in obese adipose tissue [20], multiple studies have now shown that inflammatory T cells also play a key role in driving obesity-associated inflammation. In a study of recombinase activating gene 1 knockout (Rag1$^{-/-}$) mice, in which adaptive immune cells (lymphocytes) are not present, but the macrophage compartment remains intact, the Rag1$^{-/-}$ mice fed high-fat diet showed increased adiposity and weight gain, but had decreased inflammation in the adipose tissue compared to wildtype mice [21]. In particular, there were decreased inflammatory innate immune cells present in the adipose tissue, and IFN-γ levels were decreased [21]. These data suggest that adaptive immune cells, likely T cells, significantly contribute to the inflammatory environment of the adipose tissue in obesity either by recruitment of inflammatory innate immune cells or maintenance of the inflammatory environment by cytokine production. Moreover, mouse models that knockout the T cell receptor (TCRβ$^{-/-}$), the Th1-associated transcription factor T-bet, or the Th1-associated cytokine IFN-γ have been shown to be protected against insulin resistance and diabetes when placed on high-fat diet [22,23]. This suggests that not only are T cells required for the inflammatory environment in obesity, but they are also required for the development of the systemic metabolic phenotype associated with obesity.

Adipose-derived cytokines, called adipokines, can contribute to the inflammatory environment of the adipose tissue in obesity. One such adipokine is leptin, which is secreted in proportion to adipose tissue mass and is therefore increased in the setting of obesity and decreased in undernutrition [24]. Leptin is best known for its role in the hypothalamus where leptin receptors are highly expressed and leptin signaling leads to decreased appetite and increased energy expenditure. The roles of leptin in neuroendocrine function, energy homeostasis and metabolism have been thoroughly reviewed [25–27]. Interestingly, leptin also has a critical role as an immune modulator. Leptin has been shown to affect multiple immune cell types and mediate varying effects depending on the cell type or activation status of the cell, with particularly significant and striking effects on T cell development and function [24].

To start, leptin is required for early T cell development in the thymus. Double negative, double positive and CD4 single positive, but not CD8 single positive, thymocytes were shown

to express leptin receptor [28]. Furthermore, leptin administration to leptin deficient mice rescued CD4+ T cell development, but not CD8+ T cell development [28]. In addition to the developmental requirement, leptin has also been shown to influence CD4+ T cell function and differentiation. In particular, leptin promotes CD4+ T cell differentiation into Th1 and T helper 17 (Th17) functional subsets and increases their IFN-γ and IL-17 production, respectively [29]. One mechanism by which leptin may influence CD4+ T cell function and differentiation is through cellular metabolism [29,30]. Specifically, leptin has been shown to increase expression of the glucose transporter Glut1 and increase glucose uptake and glycolytic metabolism, thereby promoting inflammatory cytokine production [29,30]. Thus, leptin modulates metabolism systemically at the level of the hypothalamus, but also at the level of the immune cell, where it can influence immune cell function.

Given the role of T cells in driving obesity-associated inflammation and insulin resistance and the role of leptin in promoting CD4+ T cell inflammatory function, we asked: what is the role of leptin signaling in driving T cell inflammation in the context of obesity-associated changes in systemic metabolism? To answer this question, we set up the following studies. T cell-specific leptin receptor conditional knockout mice and littermate controls were either fed high-fat diet (60% kcal from fat) or low-fat diet for 18 weeks, starting at weaning. Body weights and blood glucose levels were monitored throughout the study. At the completion of the dietary intervention, a glucose tolerance test was performed, and CD4+ T cells were isolated from spleens to measure cellular metabolism and function.

## Materials and methods

### Animals

T cell-specific leptin receptor (LepR) knockout mice were generated by crossing CD4Cre transgenic mice with LepR-floxed mice (Jackson Laboratory). The following genotypes were studied: *CD4Cre+LepR<sup>fl/fl</sup>* (leptin receptor knockout mice; LepRcKO) and *CD4Cre-LepR<sup>fl/fl</sup>* (controls; Ctrl). Mice were weaned at 3 weeks of age onto either low-fat, normal chow (NC, 10-kcal% fat, LabDiet, St. Louis, MO) or high-fat diet (HFD, 60-kcal% fat, Research Diets, New Brunswick, NJ) and remained on this diet for 18 weeks. Mice were group housed (up to 5 per cage), maintained at ambient temperature, and given ad libitum access to food and water. Mouse weights were collected weekly. All animal protocols were approved by the Institutional Animal Care and Use Committees at Duke University or the University of North Carolina at Chapel Hill.

### Blood glucose readings and glucose tolerance tests

Blood glucose readings were taken using a One Touch Verio glucometer and strips. Mice were fasted for 4 hours prior to measuring blood glucose. Tails were snipped and one drop of blood deposited onto glucometer strip. To measure glucose tolerance, mice were fasted for 6 hours prior to glucose tolerance test. A baseline blood glucose reading was taken from each mouse. Mice were then injected intraperitoneally with 2 g/kg body weight of glucose. Blood glucose readings were taken at 15 minutes, 30 minutes, 60 minutes, 90 minutes, and 120 minutes following injection.

### Tissue collection and processing

Method as described in Alwarawrah et al. 2020 [31]. Mice were euthanized using $CO_2$ inhalation. Spleens were mashed and strained in PBS, washed, and resuspended. A portion of splenocytes were set aside for flow cytometry and the rest were used to isolate CD4+ T cells using the

StemCell CD4$^+$ T cell isolation kit (StemCell technologies, Vancouver, BC, Canada). Isolated CD4$^+$ T cells were analyzed by extracellular flux analysis.

## Flow cytometry

Method as described in Alwarawrah et al. 2020 [31]. Treg staining: Two million splenocytes were fixed and permeabilized using the Foxp3 Transcription Factor Staining Buffer kit (eBioscience) and stained for Foxp3 following the manufacturer instructions. For the identification of Treg cells, the following antibodies were used: BV421 Armenian Hamster anti-mouse CD3e (Biolegend, San Diego, CA), BV605 rat anti-mouse CD4 (Biolegend), PeCy7 rat anti-mouse CD25 (Biolegend), AF488 rat anti-mouse Foxp3 (Biolegend). Cytokine staining: For the identification of effector T cells (Th1 and Th17) and evaluation of their function, the following antibodies were used: BV421 Armenian Hamster anti-mouse CD3e (BD BioSciences), BV605 rat anti-mouse CD4 (Biolegend), AF488 rat anti-mouse CD8a (Biolegend), APC rat anti-mouse IL-17A (Biolegend), and PE/Cy7 rat anti-mouse IFNγ (Biolegend). Five million splenocytes were stimulated for 4.5 h in complete media containing Golgi Plug (2 µg/ml) (BD Biosciences), PMA (50 ng/ml) (Sigma-Aldrich, St. Louis, MO), and ionomycin (1 µg/ml) (Sigma-Aldrich), then permeabilized and fixed with Cytofix/Cytoperm kit (BD Biosciences) and stained for IFN-γ (Biolegend) and IL-17A (Biolegend) following the manufacturer's protocol. Samples were acquired on a ThermoFisher Attune NxT flow cytometer, and data were analyzed using FlowJo (Treestar, Ashland, OR).

## Metabolic flux assays

Method as described in Alwarawrah et al. 2020 [31]. CD4$^+$ T cells were washed with Seahorse XF RPMI 1640 media (Agilent, Santa Clara, CA) and plated at a density of 250,000 cells/well (50 µL) in a Seahorse XFe96 plate (Agilent) pre-coated with Cell-Tak (Corning, Corning, NY). After spinning down the plate at 200 rpm for 1 min, the plate was incubated for 30 min in a humidified 37˚C incubator in the absence of $CO_2$. Seahorse XF RPMI 1640 media (130 µL) was added, and the plate was incubated for an additional 20 min. Oxygen consumption rate (OCR) and extracellular acidification rate (ECAR) were measured using a Seahorse XFe96 Analyzer (Agilent).

## Statistical analysis

Comparisons between groups were analyzed using t-test with Welch's correction assuming Gaussian distribution. Statistical analysis was performed using GraphPad Prism 9 (GraphPad Software, Inc., La Jolla, CA). All data was determined as significant by $p < 0.05$. Glucose tolerance test (GTT) analyses were performed in SAS 9.4 (SAS Institute Inc., Cary, NC) at a two-tailed significance level of 0.05. All available GTT measurements within the study were included in the analysis. Marginal models using generalized estimating equations (GEE) were implemented to account for the correlation between repeated measurements of GTT levels in each mouse. A GEE-type model with a normal distribution and identity link was implemented to test the association between sex and GTT levels as well as diet and GTT levels. An exchangeable working correlation structure was used along with the robust variance estimator for all models.

## Results

### Leptin receptor deficiency on T cells does not confer protection from systemic glucose intolerance in diet-induced obesity

To determine the role of leptin signaling in driving T cell inflammation in the context of obesity-associated changes in systemic metabolism, we generated T cell-specific leptin receptor

knockout mice by crossing leptin receptor floxed (*LepR$^{fl/fl}$*) animals with mice expressing a Cre recombinase transgene under the control of the CD4$^+$ promoter (*CD4Cre*) on the C57BL/6 background, as we have previously described [29,30]. These mice have a selective deletion of leptin receptor in both CD8$^+$ and CD4$^+$ T cells and in T cell subsets. Given our prior work demonstrating the effect of leptin on CD4$^+$ T cells, particularly Th1 and Th17 cells, as well as the relatively low expression of leptin receptor on CD8$^+$ T cells compared to CD4$^+$ T cells, we anticipated that this deletion would affect the metabolism and function of CD4$^+$ T cells more so than CD8$^+$ T cells [29,30].

T cell-specific leptin receptor knockout mice (*CD4Cre$^+$LepR$^{fl/fl}$*) and littermate controls expressing leptin receptor (*CD4Cre$^-$LepR$^{fl/fl}$*) were placed on high-fat diet (HFD; 60% kcal from fat) or low-fat, normal chow (NC; 10% kcal from fat) for 18 weeks, starting at weaning at 3 weeks of age. This study design allowed us to collect longitudinal data during the development of obesity. The mice on HFD gained significantly more weight than the mice on NC, but there was no difference in weight gain between T cell-specific leptin receptor knockout mice and littermate controls on the same diet (Fig 1A). These findings were consistent in both male and female mice; however, male mice on HFD showed increased weight gain at an earlier age than female mice, as well as greater weight gain overall (Fig 1B and 1C).

Following 18 weeks on diet, mice were fasted for 4 hours to allow for measurement of fasting blood glucoses. Fasting blood glucose levels of control mice on HFD were significantly elevated compared to control mice fed NC; however, these differences were not observed between leptin receptor conditional knockout mice on HFD compared to NC (Fig 2A). When these data were separated by sex, we found that control male, but not female, mice on HFD had significantly higher fasting blood glucose levels than their NC fed counterparts (Fig 2B and 2C). Again, this significant difference in weight gain between NC and HFD-fed male control mice was not observed in male mice with T cell-specific leptin receptor deficiency (Fig 2B).

Mice from each experimental group were then subjected to a glucose tolerance test. As shown in Fig 3A and 3B, mice fed HFD had significantly impaired glucose tolerance compared to mice fed NC, but there were no significant differences in glucose tolerance between T cell-specific leptin receptor knockout mice or control mice on either NC or HFD. These findings were maintained when mice were separated into male or female groups; however, male mice on HFD had more impaired glucose tolerance than female mice on HFD (Fig 3C and 3D), consistent with previous observations in the literature [32–35]. To determine if there were other changes to systemic metabolism that could influence immune cell responses in our experimental groups, we analyzed serum levels of cholesterol, triglycerides, and leptin and found no significant differences (S1 Fig).

## Obesity-induced changes in CD4$^+$ T cell oxidative metabolism depend on T cell-specific leptin receptor expression

Two days following glucose tolerance testing, mice were euthanized, and T cell metabolism and function were evaluated. Since CD4$^+$ T cell metabolism has been previously shown to be altered in obesity [31], we sought to characterize the metabolic phenotype of CD4$^+$ T cells from T cell-specific leptin receptor knockout mice versus littermate control mice on either NC or HFD. To that end, CD4$^+$ T cells were isolated from spleens and analyzed using extracellular flux analysis. CD4$^+$ T cells isolated from control mice fed HFD, but not from T cell-specific leptin receptor knockout mice fed HFD, showed significantly increased basal oxygen consumption rate (OCR, a surrogate marker for mitochondrial oxidation) compared to mice fed NC (Fig 4A). In contrast, CD4$^+$ T cells from T cell-specific leptin receptor knockout mice or

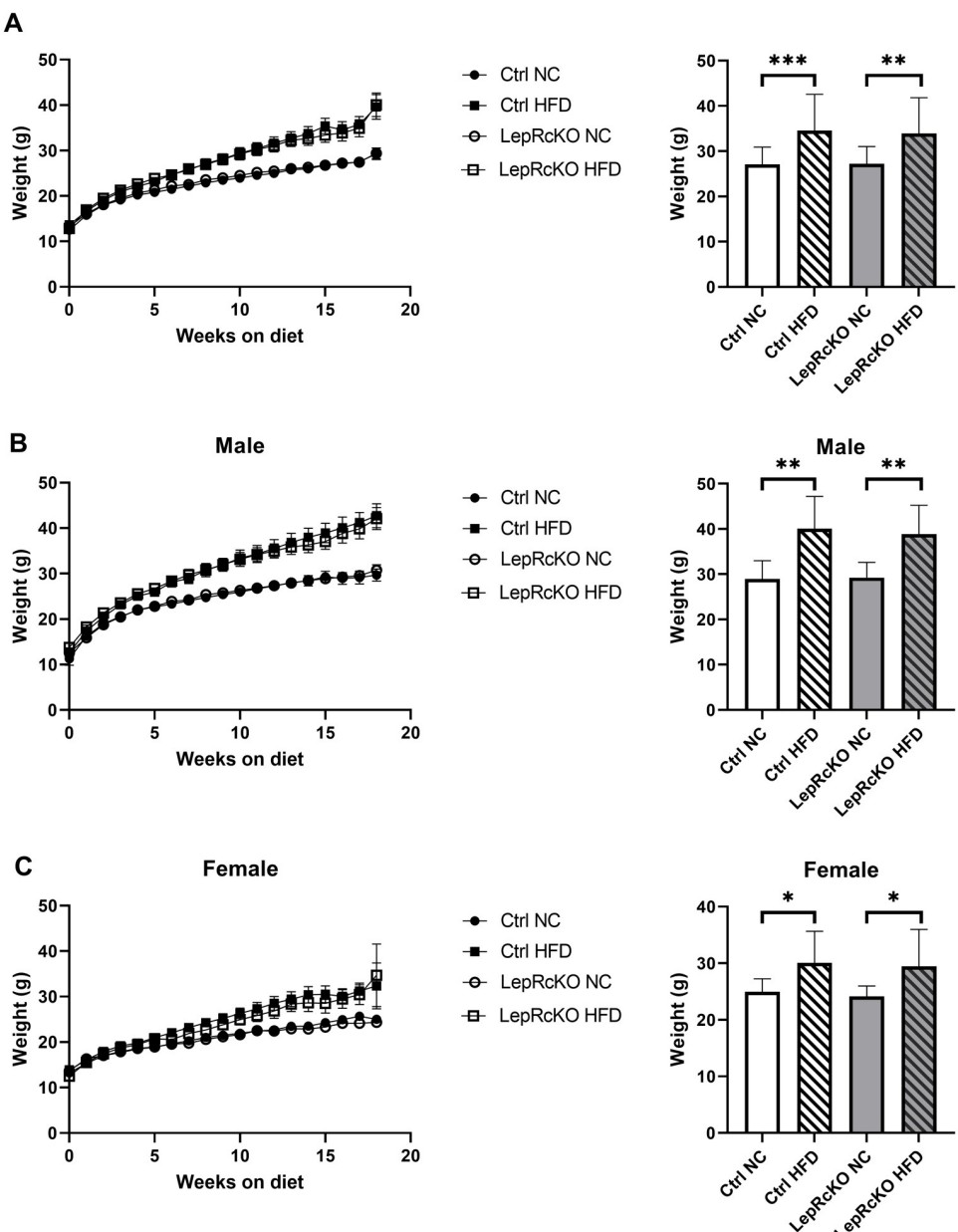

**Fig 1. T cell-specific leptin receptor knockout mice and littermate controls have equivalent weight gain following high-fat diet.** T cell-specific leptin receptor knockout mice (LepR cKO) and littermate controls (Ctrl) were fed low-fat, normal chow (NC) or high-fat diet (HFD) for 18 weeks. Body weights were measured weekly and graphed versus time. Data plotted as mean ± standard error. Bar graphs show weights at 16 weeks on diet. Data shown as mean ± standard deviation. All experimental mice (n = 18-20/group) are shown in (a), male mice (n = 9-14/group) are shown in (b), female mice (n = 9-11/group) are shown in (c) over the course of the experiment and then at 16 weeks (bar graphs). Data analyzed using student's t test with Welch's correction (*p<0.05; **p<0.01).

littermate control mice on either NC or HFD did not show any difference in basal extracellular acidification rate (ECAR, a surrogate marker for glycolytic metabolism) (Fig 4B). The OCR/ECAR ratio is a useful measure to determine what proportion of the energy being produced in the cells is coming from mitochondrial oxidation versus glycolytic metabolism. In our study, control mice on HFD had a significantly increased OCR/ECAR ratio compared to control

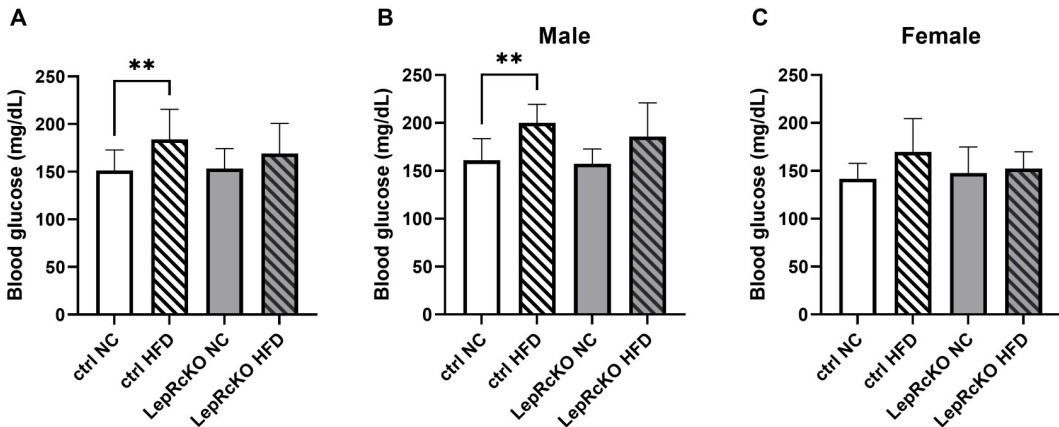

**Fig 2. Leptin receptor knockout prevents impaired fasting glucose levels in male, but not female, mice fed high-fat diet.**
T cell-specific leptin receptor knockout mice (LepR cKO) and littermate controls (Ctrl) were fed low-fat, normal chow (NC) or high-fat diet (HFD) for 18 weeks. Fasting blood glucose readings were taken after a 4 hour fast. Readings from all experimental mice (n = 12-16/group) are shown in (a), fasting blood glucose readings from male mice (n = 6-9/group) shown in (b), fasting blood glucose readings from female mice (n = 6-7/group) are shown in (c); analyzed using student's t test with Welch's correction (**p<0.01).

mice on NC (Fig 4C). However, there was no difference in OCR/ECAR ratio observed between T cell-specific leptin receptor knockout mice on NC or HFD (Fig 4C). These results suggest that leptin signaling is required, at least in part, for changes seen in T cell oxidative metabolism in obesity.

## Obesity-induced changes in CD4$^+$, but not CD8$^+$, T cell function require T cell-specific leptin receptor expression

To investigate T cell function in the experimental groups, splenocytes were analyzed by intracellular flow cytometry to determine the proportion of cells producing the pro-inflammatory cytokines IFN-γ and IL-17 or expressing the Treg-associated transcription factor Foxp3. Overall, control mice fed HFD had significantly increased proportion of IFN-γ producing CD4$^+$ T cells than control mice on NC (Fig 5A). However, no difference was observed in IFN-γ producing CD4$^+$ T cells between T cell-specific leptin receptor conditional knockout mice on NC versus HFD (Fig 5A). Interestingly, both T cell-specific leptin receptor conditional knockout and littermate control mice on HFD had significantly increased proportions of IFN-γ producing CD8$^+$ T cells when compared to mice on NC (Fig 5B). One possible explanation for the difference seen in IFN-γ production by CD4$^+$ versus CD8$^+$ T cells in the knockout mouse may be that CD8$^+$ T cells are less leptin responsive than CD4$^+$ T cells. This is consistent with findings demonstrating reduced leptin receptor expression on CD8$^+$ T cells as compared to CD4$^+$ T cells. Thus, obesity-induced changes in CD4$^+$, but not CD8$^+$, T cell production of IFN-γ depends on T cell-specific leptin receptor expression.

We found no statistically significant difference in the proportion of CD4$^+$ T cells producing IL-17 in mice fed HFD compared to mice fed NC in both T cell-specific leptin receptor knockout and littermate control mice (Fig 5C). Lastly, we examined changes in Treg cell numbers, as defined by the proportion of CD4$^+$ T cells expressing the transcription factor Foxp3 and the activation marker CD25. The proportion of Treg cells was increased in control mice, but not T cell-specific leptin receptor knockout mice, fed HFD compared to mice fed NC (Fig 5D). This result suggests that increased Treg cell proportions in obesity depend at least partially on leptin

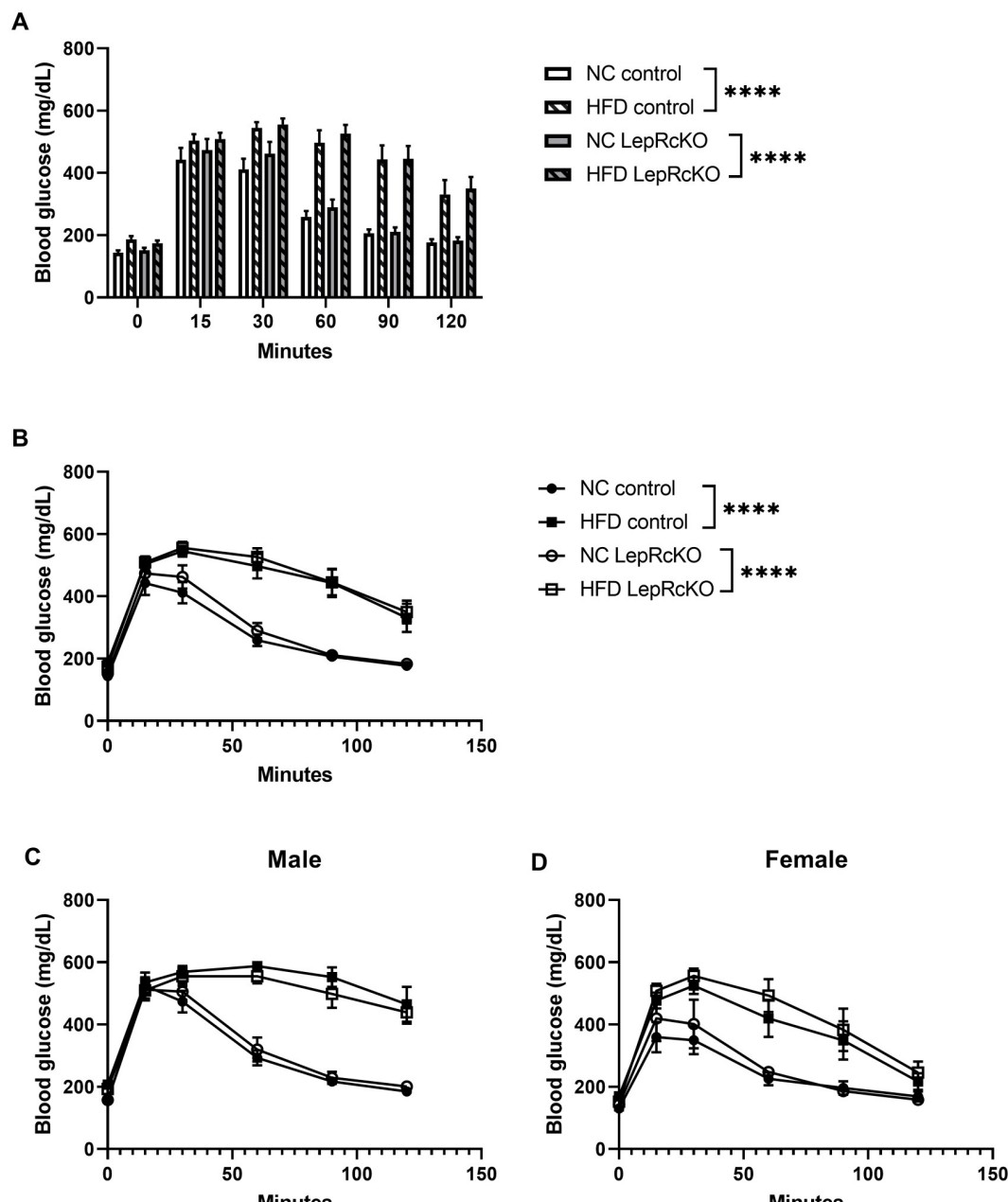

**Fig 3. T cell-specific leptin receptor knockout mice have similar glucose tolerance to littermate control mice following high-fat diet.** Glucose tolerance test (GTT) was conducted after 18 weeks on low-fat, normal chow (NC) or high-fat diet (HFD). Mice were fasted for 6 hours; baseline blood glucose measured by tail vein bleed (0-minute timepoint). 2 g/kg glucose was injected intraperitoneally, and blood glucose (mg/dL) measured at 15, 30, 60, 90, 120 minutes post-injection. Bar graph shows all experimental mice (n = 10-13/group) mean ± standard error (a), Line graph shows all experimental mice (n = 10-13/group) mean ± standard error (b), male mice (n = 5-7/group) mean ± standard error shown in (c), female mice (n = 5-7/group) mean ± standard error shown in (d); analyzed using generalized estimating equations (GEE) model (****$p < 0.0001$).

receptor expression and signaling. Lastly, we performed multiplex analysis to compare serum cytokine levels in our experimental groups, and we found no significant differences in circulating levels of TNF, IFN-γ, IL-17, IL-1β, and IL-6 (S3 Fig). Altogether, these findings demonstrate that T cell-specific leptin signaling is required for some of the changes in CD4[+] T cell

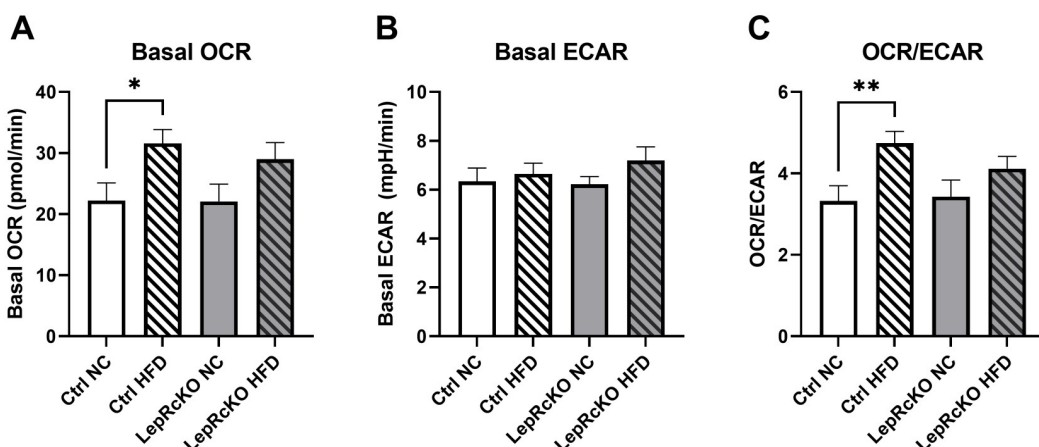

**Fig 4. T cell-specific leptin signaling is required for changes in CD4+ T cell metabolism observed in mice fed high-fat diet compared to low fat diet.** T cell-specific leptin receptor knockout mice (LepR cKO) and littermate controls (Ctrl) were fed low-fat, normal chow (NC) or high-fat diet (HFD) for 18 weeks. CD4+ T cells were isolated from spleens and extracellular flux analysis was performed. Basal oxygen consumption rate (OCR) (mean ± standard error) shown in (a), basal extracellular acidification rate (ECAR) (mean ± standard error) shown in (b), OCR/ECAR ratio (mean ±standard error) shown in (c). Data analyzed using student's t test with Welch's correction (*p<0.05; **p<0.01; n = 15-19/ experimental group).

metabolism and function observed in obesity but has minimal effects on obesity-associated systemic metabolism.

## Discussion

As an increasing proportion of the United States population is diagnosed with obesity, it is becoming critically important to understand the implications of obesity for the health of those individuals. Studying the immune response is an important aspect of this, as obesity-associated changes in immune cells have been shown to drive inflammation which promotes systemic metabolic disease [9]. Specifically, it has been demonstrated that T cells have a critical role in driving obesity-associated inflammation leading to systemic insulin resistance [21]. We have previously shown that leptin is a key regulator of both CD4+ T cell metabolism and effector cell differentiation and function [29,30]. Given the role of T cells in driving obesity-associated inflammation and insulin resistance and the role of leptin in promoting CD4+ T cell inflammatory function, we set out to determine whether leptin signaling in T cells was required for the development of glucose intolerance and insulin resistance in obesity.

Our conditional leptin receptor knockout mouse, which has leptin receptor selectively deleted in T cells, allowed us to interrogate whether leptin signaling to T cells could be promoting T cell metabolic changes, increased inflammatory cytokine production, and subsequent systemic metabolic disease in the context of obesity. We have previously reported metabolic differences between T cells from C57BL/6 mice fed NC versus HFD [31]. The data shown here support our previous work showing that CD4+ T cells from diet-induced obese control mice have increased oxidative metabolism (OCR), as well as an increased ratio of oxidative to glycolytic metabolism (OCR:ECAR) compared to CD4+ T cells from lean control mice. However, CD4+ T cells from HFD versus NC-fed T cell-specific leptin receptor knockout mice did not show the same metabolic changes. These results suggest that direct leptin signaling mediates metabolic remodeling of CD4+ T cells in obesity resulting in significantly increased basal oxidative metabolism and an increase in the ratio of oxidative to glycolytic metabolism characteristic of T cells from obese mice.

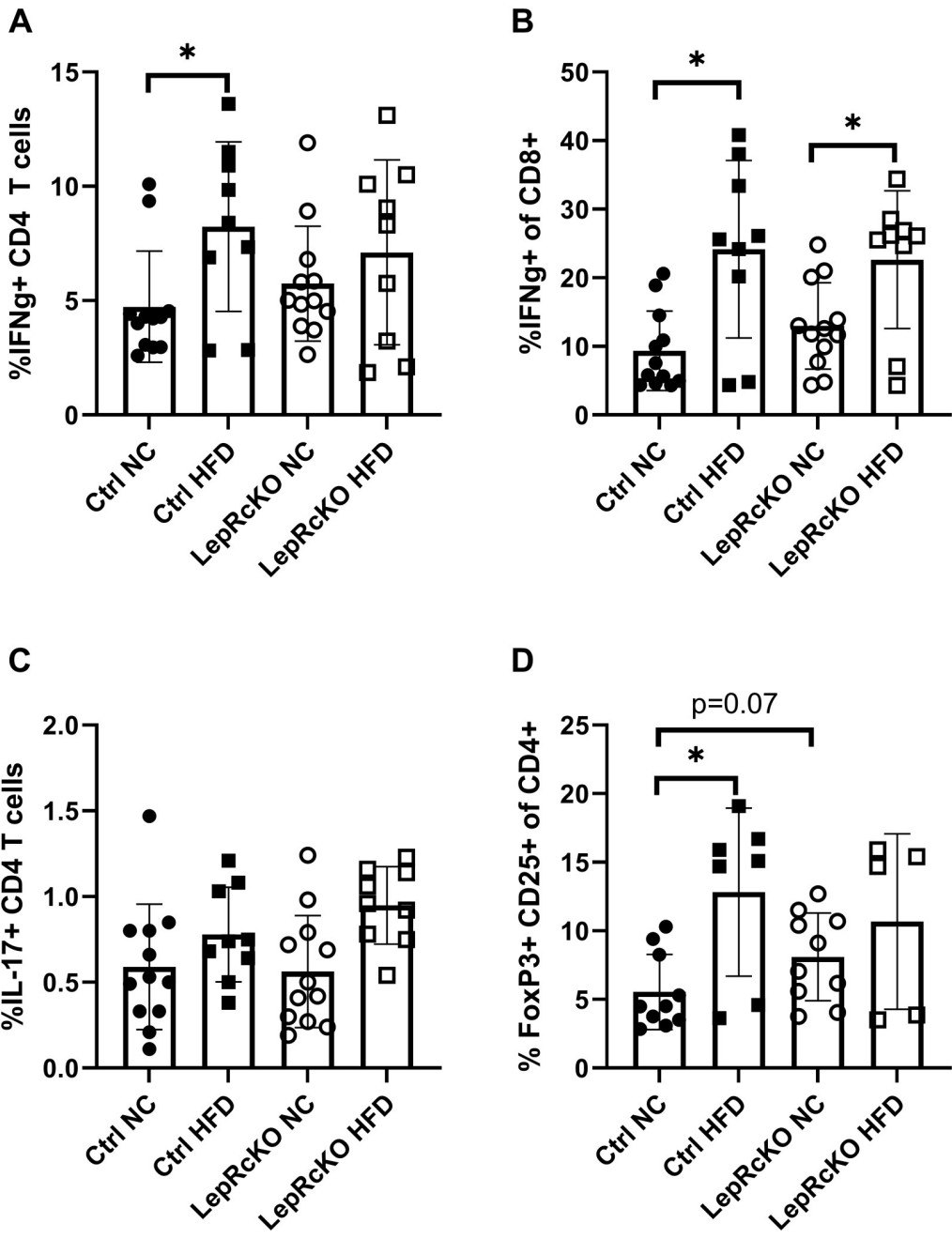

**Fig 5. Splenic CD4$^+$ T cells isolated from leptin receptor knockout mice are altered following high-fat diet compared to CD4$^+$ T cells isolated from littermate controls.** T cell-specific leptin receptor knockout mice (LepR cKO) and littermate controls (Ctrl) were fed low-fat, normal chow (NC) or high-fat diet (HFD) for 18 weeks. CD4$^+$ T cells were isolated from spleens and cytokine production or Foxp3 transcription factor was measured by intracellular flow cytometry. Proportion of splenic CD4$^+$ T cells that produce IFN-γ (a), Proportion of splenic CD8$^+$ T cells that produce IFN-γ (b), Proportion of splenic CD4$^+$ T cells that produce IL-17 (c), Proportion of splenic CD4$^+$ T cells that produce CD25 and Foxp3 (Treg cells) (d); analyzed using student's t test (*p<0.05) with Welch's correction. Each plotted point represents one mouse; n = 9–12/experimental group.

We also found differences when we interrogated the proportions of splenic T cells from our experimental groups. An increase in the proportion of IFN-γ producing CD4[+] T cells was observed in control mice fed HFD compared to NC, which was not observed in CD4[+] T cells from T cell-specific leptin receptor knockout mice. On the other hand, the proportion of IFN-γ producing CD8[+] T cells was increased following HFD compared to NC in both T cell-specific leptin receptor knockout mice and littermate controls. This differing result in CD4[+] versus CD8[+] T cells is consistent with the fact that leptin receptor is expressed at lower levels on CD8[+] T cells than CD4[+] T cells. Therefore, CD8[+] T cells may retain their IFN-γ producing phenotype in the context of HFD-induced obesity, even when leptin receptor is deleted.

To our surprise, deletion of leptin signaling in T cells had a mild influence on fasting blood glucose in obesity but did not significantly affect systemic metabolic disease. Glucose tolerance tests between HFD-induced obese T cell-specific leptin receptor knockout mice and littermate controls did not show any significant differences. One possible explanation for these negative results is a redundancy in signaling pathways downstream of leptin as well as other nutritionally regulated hormones that are increased in the setting of obesity. For example, IL-6 is known to be increased in obesity, shares signaling pathways with leptin, and can have synergistic effects on T cells. In the absence of leptin signaling in obesity, other cytokine signals may be compensatory.

One limitation of our study is that we only investigated the role of leptin receptor signaling directly on T cells. While T cells have been shown to be important for obesity-associated inflammation, there are other immune cells that have been shown to have a role in this process. In particular, macrophages are known to be important drivers of inflammation in the adipose tissue in the setting of obesity. To this point, several studies have shown that reducing macrophage recruitment to or eliminating macrophages from the adipose tissue improves insulin sensitivity [36–40]. Furthermore, IL-1 receptor I deficient mice have improved glucose tolerance and reduced inflammation in obesity, suggesting that macrophage production of IL-1β is a critical inflammatory driver in obesity [39]. A study investigating the role of leptin signaling in myeloid cells found that specific deletion of leptin receptor in myeloid cells did not significantly affect glycemic control [41]. However, selective reconstitution of leptin receptor in myeloid cells in mice that were otherwise leptin receptor deficient resulted in improved glucose tolerance. This combination of results suggests that there is a complex signaling network involving immune cells, adipose tissue, and soluble mediators such as leptin and other nutritionally regulated cytokines. Future studies should investigate the roles of other immune cells, endocrine signals, and inflammatory cytokines in driving obesity-associated inflammation and glucose intolerance.

In this study, we set out to understand whether the leptin signal to T cells in obesity is sufficient to drive systemic inflammation and metabolic disease. Using our T cell-specific leptin receptor conditional knockout mouse, we interrogated the role of leptin signaling on the pathogenesis of obesity as driven by the T cell. We found that loss of leptin signaling to T cells was insufficient to influence glucose tolerance in obesity, although leptin deficiency did prevent impaired fasting glucose levels compared to control mice in the HFD setting, in a sex dependent manner. Despite these minimal effects on systemic metabolism, T cells isolated from leptin receptor knockout mice on HFD lacked some of the metabolic and functional changes observed in T cells from control mice on HFD. These results contribute to a body of work that aims to understand the signals influencing immune cell function and inflammation in obesity. Understanding immune cells in obesity is critical for the development of therapeutics to ameliorate the consequences of obesity and subsequent metabolic disease. With an increasing proportion of individuals with obesity in both the United States and around the world, treating

the inflammation and downstream consequences of obesity will become an increasing priority.

## Supporting information

**S1 Fig. T cell-specific leptin receptor knockout mice have similar serum cholesterol, triglyceride, and leptin levels compared to littermate control mice following high-fat diet.** Mouse serum was collected after 18 weeks on low-fat, normal chow (NC) or high-fat diet (HFD) and analyzed for concentration of (A) total cholesterol and (B) triglycerides using clinical chemistry tests performed by Animal Clinical Laboratory Services core facility at University of North Carolina at Chapel Hill, and (C) leptin levels by serum ELISA (ThermoFisher). One way ANOVA corrected for multiple comparisons. No significant differences were observed between the groups. n = 5–7 mice/group.
(TIF)

**S2 Fig. Splenic CD4$^+$ T cells isolated from leptin receptor knockout mice are altered following high-fat diet compared to CD4$^+$ T cells isolated from littermate controls.** Representative flow cytometry plots with gating are shown from each group showing cytokine production for CD4$^+$ T cells (top row), CD8$^+$ T cells (middle row), and expression of FoxP3 on CD4$^+$ T cells (bottom row). (A) Pre-gated on lymphocytes, single cells, CD3$^+$, CD4$^+$; percentage IL-17$^+$ and percentage IFN-$\gamma^+$ shown (B) Pre-gated on lymphocytes, single cells, CD3$^+$, CD8$^+$ (C) Pre-gated on lymphocytes, single cells, CD3$^+$, CD4$^+$; percentage FoxP3$^+$CD25$^+$ shown.
(TIF)

**S3 Fig. T cell-specific leptin receptor knockout mice have similar levels of circulating cytokines compared to littermate control mice following high-fat diet.** Multiplex analysis of cytokine levels was performed on selected mouse serum samples using Bio-Rad Bio-Plex Pro Th17 multiplex cytokine analysis assay to examine levels of (A) TNF, (B) IFN-$\gamma$, (C) IL-17, (D) IL-1$\beta$, and (E) IL-6. IL-10 was also analyzed but levels were below the limit of detection.
n = 5–7 mice/group except where below limit of detection, in which case sample was excluded from graph. One way ANOVA corrected for multiple comparisons. No significant differences were observed between groups.
(TIF)

**S1 Data.**
(XLSX)

## Author Contributions

**Conceptualization:** Nancie J. MacIver.

**Data curation:** Kaitlin Kiernan, Amanda G. Nichols, Yazan Alwarawrah.

**Formal analysis:** Kaitlin Kiernan.

**Investigation:** Kaitlin Kiernan, Amanda G. Nichols, Yazan Alwarawrah.

**Methodology:** Kaitlin Kiernan, Nancie J. MacIver.

**Supervision:** Nancie J. MacIver.

**Writing – original draft:** Kaitlin Kiernan.

**Writing – review & editing:** Nancie J. MacIver.

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
