## [Decision Letter · Decision Letter 0]

17 Mar 2023

PONE-D-23-03100

Effects of T cell leptin signaling on systemic glucose tolerance and T cell responses in obesity

PLOS ONE

Dear Dr. MacIver,

Thank you for submitting your manuscript to PLOS ONE. After careful consideration, we feel that it has merit but does not fully meet PLOS ONE’s publication criteria as it currently stands. Therefore, we invite you to submit a revised version of the manuscript that addresses the points raised during the review process.

We look forward to receiving your revised manuscript.

Kind regards,

Sadiq Umar

Academic Editor

PLOS ONE

Journal Requirements:

    "This study was supported by NIH R01-DK106090 (NJM)"

    "This study was supported by NIH R01-DK106090 (NJM)

Reviewers' comments:

Reviewer's Responses to Questions

**Comments to the Author**

1. Is the manuscript technically sound, and do the data support the conclusions?

Reviewer #1: Yes

Reviewer #2: Yes

2. Has the statistical analysis been performed appropriately and rigorously? 

Reviewer #1: Yes

Reviewer #2: Yes

3. Have the authors made all data underlying the findings in their manuscript fully available?

Reviewer #1: Yes

Reviewer #2: Yes

4. Is the manuscript presented in an intelligible fashion and written in standard English?

Reviewer #1: No

Reviewer #2: Yes

5. Review Comments to the Author

Reviewer #1: - Authors need to revise the structure of results section. Please structure with proper subsections and describe your results separately; not only with Figures legends/captions. Also, figure legends should be submitted in a different section or along with figures.

- Why you didn't check lipid profile? What about systemic effects of hypercholesterolimia? It greatly trigger cardiovascular and immune response. Please justify.

- You could also take pictures of spleen from each mice. Could help justify splenomegaly!

- Study could elaborate several other factors and parameters of systemic inflammation. Why didn't you do that. Please justify other possible limitations.

Reviewer #2: In this article, the authors have focused on understanding the leptin signaling pathway driving T-cell inflammation in obesity. This is a well-written manuscript with minor correction that needs to be addressed,

Line 61 Expand Rag-/-

Quote reference for all the protocols followed in the method section

Line 148 quote the company

Include Scatter plot for flow cytometry

Give a line of reason for every result stated in the study. For example, why are the metabolism and function affected in CD4+ T cells when compared to CD8+ T cells?

Check for grammatical mistakes throughout the manuscript

6. PLOS authors have the option to publish the peer review history of their article (what does this mean?). If published, this will include your full peer review and any attached files.

Reviewer #1: No

Reviewer #2: **Yes: **FARHATH SULTANA

---

## [Author Response · Author response to Decision Letter 0]

15 May 2023

Response to reviewers

We thank the reviewers for their careful review of our manuscript. We have addressed all reviewer comments below and now present an improved manuscript for review. 

Reviewer #1:

1. Authors need to revise the structure of results section. Please structure with proper subsections and describe your results separately; not only with Figures legends/captions. Also, figure legends should be submitted in a different section or along with figures.

Thank you for this suggestion. We agree that adding subsections would improve the structure of the manuscript. In response to this comment, subsections were added to the results section. Figure legends are placed in the text as per PLOS manuscript guidelines. 

2. Why you didn't check lipid profile? What about systemic effects of hypercholesterolemia? It greatly trigger cardiovascular and immune response. Please justify.

Thank you for this comment. We agree that systemic effects of hypercholesterolemia could be impacted in our mouse model and subsequently influence T cell responses. In response to this comment, we performed lipid analyses on serum samples from this study. Please see Supplemental Figure 2. We did not see any significant differences in total cholesterol or triglyceride levels between the experimental groups.

3. You could also take pictures of spleen from each mice. Could help justify splenomegaly!

While we did not have photographs of the spleens from these mice, we did record spleen weights and weight of visceral adipose tissue, so we have included that information in the response to reviewers attached file and in the revised cover page. 

4. Study could elaborate several other factors and parameters of systemic inflammation. Why didn't you do that. Please justify other possible limitations.

Thank you for this comment. We agree that more markers and parameters of systemic inflammation would be informative. In response to this comment, we performed multiplex analysis on serum samples from our mouse cohort. We assessed serum levels of TNF, IFN-γ, IL-17, IL-1β, and IL-6. We did not find any significant differences in serum cytokine levels between the groups. We also performed a leptin ELISA to measure serum leptin levels in these mice. This data is included in Supplemental Figures 1 and 3.

Reviewer #2: 

1. In this article, the authors have focused on understanding the leptin signaling pathway driving T-cell inflammation in obesity. This is a well-written manuscript with minor correction that needs to be addressed,

Line 61 Expand Rag-/-

Quote reference for all the protocols followed in the method section

Line 148 quote the company

Include Scatter plot for flow cytometry

Give a line of reason for every result stated in the study. For example, why are the metabolism and function affected in CD4+ T cells when compared to CD8+ T cells?

Check for grammatical mistakes throughout the manuscript

We thank the reviewer for these helpful comments. In response to the comments from this reviewer, changes to the text were made, including expanding on “Rag-/-” to clarify, referencing our publications where methods have been published previously, and including several sentences explaining the results presented. We also included flow cytometry scatter plots in Supplemental Figure 2.

---

## [Editor Report · Decision Letter 1]

17 May 2023

Effects of T cell leptin signaling on systemic glucose tolerance and T cell responses in obesity

PONE-D-23-03100R1

Dear Dr. MacIver,

We’re pleased to inform you that your manuscript has been judged scientifically suitable for publication and will be formally accepted for publication once it meets all outstanding technical requirements.

Kind regards,

Sadiq Umar

Academic Editor

PLOS ONE

Additional Editor Comments (optional):

Recommendation for acceptance.
---

## [Editor Report · Acceptance letter]

24 May 2023

PONE-D-23-03100R1 

Effects of T cell leptin signaling on systemic glucose tolerance and T cell responses in obesity 

Dear Dr. MacIver:

I'm pleased to inform you that your manuscript has been deemed suitable for publication in PLOS ONE. Congratulations! Your manuscript is now with our production department. 

Kind regards, 

on behalf of

Dr. Sadiq Umar 

Academic Editor

PLOS ONE